# Comparative Meta-Analysis of the Effect of Concentrated, Hydrolyzed, and Isolated Whey Protein Supplementation on Body Composition of Physical Activity Practitioners

**DOI:** 10.3390/nu11092047

**Published:** 2019-09-02

**Authors:** Luis Henrique A. Castro, Flávio Henrique S. de Araújo, Mi Ye M. Olimpio, Raquel B. de B. Primo, Thiago T. Pereira, Luiz Augusto F. Lopes, Erasmo B. S. de M. Trindade, Ricardo Fernandes, Silvia A. Oesterreich

**Affiliations:** 1Graduate Program in Health Sciences—Federal University of Grande Dourados (UFGD), Dourados 79804-970, Brazil; 2Faculty of Health Sciences—Federal University of Grande Dourados/Universitary Hospital of Federal University of Grande Dourados, Dourados 79823-501, Brazil; 3Graduate Program in Nutrition—Federal University of Santa Catarina (UFSC), Santa Catarina 88040-970, Brazil; 4Graduate Program in Food, Nutrition and Health—Federal University of Grande Dourados (UFGD), Dourados 79804-970, Brazil

**Keywords:** whey proteins, exercise, sports, resistance training, fat-free mass, fat mass, systematic review

## Abstract

Whey protein (WP) is a dairy food supplement and, due to its effects on fat-free mass (FFM) gain and fat mass (FM) loss, it has been widely consumed by resistance training practitioners. This review analyzed the impact of WP supplementation in its concentrated (WPC), hydrolyzed (WPH) and isolated (WPI) forms, comparing it exclusively to isocaloric placebos. Random effect meta-analyses were performed from the final and initial body composition values of 246 healthy athletes undergoing 64.5 ± 15.3 days of training in eight randomized clinical trials (RCT) collected systematically from five scientific databases. The weighted mean difference (WMD) was statistically significant for FM loss (WMD = −0.96, 95% CI = −1.37, −0.55, *p* < 0.001) and, in the analysis of subgroups, this effect was maintained for the WPC (WMD = −0.63, 95% CI = −1.19, −0.06, *p* = 0.030), with protein content between 51% and 80% (WMD = −1.53; 95% CI = −2.13, −0.93, *p* < 0.001), and only for regular physical activity practitioners (WMD = −0.95; 95% CI = −1.70, −0.19, *p* = 0.014). There was no significant effect on FFM in any of the scenarios investigated (*p* > 0.05). Due to several and important limitations, more detailed analyses are required regarding FFM gain.

## 1. Introduction

Derived from the manufacture of cheese and other dairy products, whey protein (WP) is a supplementary food widely used by food and pharmaceutical industries due to its nutritional value with emphasis on the presence of high levels of β-lactoglobulin, α-lactalbumin, immunoglobulins, lactoferrin, lactose, minerals, vitamins, and lipids [1,2,3].

Whey protein has high concentration of essential amino acids such as leucine [4] and, in addition, its serum proteins are also source of bioactive peptides that can act in the treatment of clinical outcomes such as cancer, HIV infection, hepatitis B, cardiovascular diseases, and osteoporosis [2,3].

Among the manufacturing techniques used for the serum proteins isolation, membrane filtration stands out, which can act in five processes: microfiltration, ultrafiltration, nanofiltration, reverse osmosis, and ion-exchange chromatography [5]. After this fractionation process, the three most consumed types of whey protein originate: isolated (WPI), concentrated (WPC), and hydrolyzed (WPH) [6]. Its stratification occurs, among other factors, by the final protein content on a scale of about ≤90% for WPI and ≥89% for WPC [6,7]. On the other hand, WPH reaches variable content depending on the efficacy of the polypeptide chain enzymatic breakdown influenced by environmental conditions such as temperature and pH [8].

Whey supplementation has been performed by physical activity practitioners, especially by those who practice strength and/or resistance training [9]. In 2013, for example, the American production of WPI and WPC exceeded 1.2 million tons [6]. This fact is mostly explained by the wide dissemination of its benefits for various outcomes linked to muscle gains, such as increase in strength and resistance, fat-free mass (FFM) and lean mass (LM) gain, and fat mass (FM) loss [9,10].

Conceptually, LM is constituted by the smooth and striated skeletal muscle portion (besides bone mass, visceral, tissue and their corresponding aqueous fractions) while FFM, both commonly increased in resistance exercise practitioners, represents the entire human body except, in this case, the visceral and tissue adipocytic aggregate, (FM) [11,12,13].

Although the positive effects of the dose/response relationship between WP supplementation and improvement of body composition are already known, factors such as level, type of physical activity and trainability aspects have strong influence on individual final results [11]. Moreover, it is necessary to consider that the maximum saturation dose for the plateau is still uncertain, which makes establishing prevalence values for the risks associated with its continuous long-term ingestion impossible [2,4,6].

Although it is a consensus that, compared to the general population, the daily protein recommendation dose for athletes is higher than that estimated by the Dietary Reference Intakes (DRI) [14], its exact values are frequently reviewed. Each year, new predictions for different sports are proposed, based on primary studies conducted with physically active specific population groups [15,16].

In addition to the type of mechanical effort performed, another determining factor for the improvement of anthropometric markers is the food choice for this population. Criteria such as origin, quality, digestibility, purity, and protein content of the diet, for example, are directly related to the type of supplemented protein [10,11,15,16,17]. Moreover, it has been established that factors such as the content of serum solids, chemical and protein composition, functional properties, and mechanisms of functionality (solubility, gelling, foam stability, hydration, diffusion, adsorption, among others) are sensibly different between WPC and WPI [18]. Adequate knowledge of these determinants, therefore, is critical for allowing adequate choice among the available variants of this product.

However, even with distinct relevance for the clinical sport practice, when addressing these factors, our research group—after an extensive search in the leading scientific databases—did not find any systematic reviews and/or meta-analyses that have investigated the comparative relationship between types of WP and comparable placebos.

Thus, to our knowledge, this is the first study to perform a metanalytic quantification of the effects of WPI, WPC, and WPH supplementation compared to isocaloric carbohydrates in order to evaluate which of these variants is the most effective for body composition improvement in healthy adult physical activity practitioners.

## 2. Materials and Methods

This study was previously registered in the International Prospective Register of Systematic Reviews (PROSPERO) under protocol CRD42019121382 and conducted in conformity with recommendations of Preferred Reporting Items for Systematic Review and Meta-analysis (PRISMA) [19]. It was also registered and approved by the Research Committee and by the Board of Directors of the Faculty of Health Sciences—Federal University of Grande Dourados (protocol No. 054/2019 and 130/2019, respectively).

All methodological steps were supervised by two authors of this paper and performed independently by at least two other authors, who did not maintain contact with each other during the execution of each process. The judgment of a third author was used in all cases of divergence. For further details, see the Authors Contribution section.

### 2.1. Eligibility Criteria and Characterization of Intervention

Randomized Clinical Trials (RCT) were collected (1), published in the English language (2), researched individuals were, at the moment of acceptance, healthy (3), aged 18–60 years (4), and non-sedentary (5).

At least two interventions should be present: WPC, WPH, and/or WPI supplementation (6), and strength and/or physical resistance activity (7). Crossover studies were included as long as there is total exchange of participants between groups and the occurrence of sufficient time interval (minimum of 20% of the total protocol duration) among experiment rounds (8).

For this review, WP supplementation was defined as the voluntary, periodic, and regular consumption of this product, of adjuvant nature to the routine diet, in order to achieve increases in skeletal muscle mass gain by physical activity practitioners, however, uses such as therapeutic, prophylactic, and alternative were not included [15]. As previously established in the PROSPERO protocol, the cutoff point for the implemented dose parameters and the periodicity of WP use was defined according to criteria adopted by each RCT included in this review. Subsequently, this assessment was considered in the delimitation of categories adopted in the data extraction stage.

The confirmation of the WP supplemented typology was provided by the identification in the text itself (when explicit) and by the analysis of the product label on the manufacturer’s website (when the brand was mentioned). In other cases, information was requested from the authors.

### 2.2. Search Strategy and Preliminary Screening

The central search strategy was performed online in 5 databases: PubMed, Cochrane (Central), Scopus, SPORTdiscus, and Web of Science. In addition, a parallel verification of completed and unpublished studies was conducted at the National Library of Medicine (NIH, Bethesda, MD, USA), the network of Brazilian Clinical Trials Registries (REBEC), and the International Clinical Trials Registry Platform from the World Health Organization (WHO/ICTRP).

For published studies, the strategy adopted was based on the PICO structure (Population, Intervention, Comparator, and Outcome), adopting related keywords and most common synonyms (Table 1). For unpublished studies, the search was designed more broadly, being composed only of the search engine “whey protein.”

In both cases, descriptors were chosen from results of preliminary searches under the critical evaluation of physical education professionals who were members of this team of reviewers. The literature of systematic reviews in the same area was also consulted, and the final criterion for the selection of keywords was sensitivity: more wide-ranging strategies were always preferred over more specific searches.

There was no limit for year of publication, and strategies were adapted to standards of each database regarding the use of filters, Boolean operators, and wildcards. The selection filters for the comparator; however, had preference over the application of keywords in cases allowed by the database (see Appendix A).

This step was performed independently in three different access networks between December 10 and 14, 2018, and the search that returned the grossest results was chosen for screening. In addition, eligible studies of the bibliographic reference of these results were also included (for further information see Appendix A).

In order to ensure efficiency and accuracy of the process, after analyzing duplicates, screening was performed in three sequential and cumulative steps, each with its own checkpoints, respectively: title (for criteria 1 and 2), abstract (for 3, 4 and 5), and full text (6, 7 and 8) using the EndNote^®^ X7 citation manager (Philadelphia, PA, USA).

However, confirming our initial hypothesis based on preliminary searches, high degree of methodological variability was found in the resulting RCT regarding both protein supplementation and physical activity, which would invariably make this review impossible.

Thus, as previously established in the PROSPERO protocol, new inclusion and exclusion criteria were adopted in order to examine which measurement methods were more prevalent and more homogeneously used in the evaluation of skeletal muscle gain regarding whey protein supplementation, in addition to other criteria that seemed necessary (see Section 2.3).

### 2.3. Data Extraction and Final Screening

Three RCTs discarded from the screening were randomly chosen, and a preliminary spreadsheet containing the main fields provided for both parametric and non-parametric data already categorized was structured. The extraction started only when both authors responsible for this step presented equivalent results.

Data were collected regarding type of study, methodology used in the research, details of intervention and control, recruitment process; sociodemographic characteristics of the study population, when available, possible losses of individuals, results obtained, as well as measurement time and method, criteria for inclusion and exclusion of participants, information that allowed the analysis of bias risk and effect measurements determined in addition to the method selected for their measurement.

After independent and qualitative analysis of this extraction, the following new evaluative and selective parameters were defined: among the many outcomes correlated to hypertrophy used by RCTs, body composition was selected, being the same quoted here, specifically, as the relative amount of LM, FFM, and/or FM (kg), and these values must not differ significantly (*p* < 0.05) between groups at baseline (9).

Therefore, only gold standard methods (10) such as dual energy X-ray absorptiometry (DXA), plethysmography, hydrostatic diving, and ultrasound were included [20,21]. Secondary prediction methods such as skin folds and body circumferences were excluded due to the high standardization variability for this technique according to the country of origin and due to the inherent evaluator bias [12,20,21].

Although most authors in the sports field often choose to express body composition data as LM, several studies point out that, conceptually, tissue analysis by DXA exam evaluates parameters truly related to FFM and not the lean mass itself [22,23,24,25,26,27]. In fact, LM by DXA compares to FFM except for bone minerals [22] and, in dissection, its estimated values for lipid-free skeletal muscles and skin are centesimally equivalent [27]. Therefore, the specific DXA data of the studies included in this review were categorized into values related to FFM.

Aiming at obtaining higher comparative power, homogeneity of experimental designs and advancement in relation data described in literature, only RCTs that contained at least one group supplemented with WP were selected, without associations, and this group was compared to at least one placebo (11) in isocaloric condition compared to the previous one (12).

Consequently, studies aimed at analyzing parallel interventions (13) were excluded, for example, the use of artificial heat since in these cases, all groups were involved with the same type of supplementation, making comparative examination impossible.

Resulting studies with possible multiple publications were analyzed according to the following criteria: authors and year of publication, study site and clinical scenario, intervention details such as dose, frequency and time, number of participants and baseline sample characteristics and recruitment method and study duration. In cases in which multiple publications were confirmed, only the article published first was selected (14) since the sample was the same and, consequently, body composition values did not differ among studies.

For greater methodological rigor, criteria 1 to 8 were again checked in the resulting RCTs based on the full text, and then definitive screening (9–14) was performed (Figure 1). All data missing in the text and/or supplementary material were requested, on multiple occasions, from corresponding authors.

### 2.4. Risk of Bias and Quality Assessment

The qualitative evaluation of RCTs was performed following the most recent update of the Cochrane bias risk analysis instrument: the ROB 2.0 tool for randomized clinical trials and its additional version for crossover studies which, although not yet published for formatting reasons, was made public by the platform in October 2018 [28]. Randomly, 10 RCT discarded from the final screening were sent to the authors responsible for this step, and only when they returned equivalent results, this evaluation started.

All aspects predicted by the updated tool were analyzed: bias arising from the randomization process, bias due to deviations from intended interventions, due to missing outcome data, to the outcome measurement, and bias due to the selection of reported results. In all domains, predefined signaling questions were observed, pointing to each one of them, the answer corresponding to the judgment of evaluators, according to the analysis suggestion flowchart.

At the end of the process, as recommended by the guide, each study was classified into one of three overall judgment categories: low risk of bias (if all domains were as such), of some concern (if found in at least one domain, and if no high risk was identified), or high risk of bias (if found in at least one domain).

After the conclusions of this review were finalized, the analysis of quality of the generated evidence was implemented using the Grading of Recommendations, Assessment, Development, and Evaluation tool (GRADE) [29]. Parallel evidence from three similar meta-analyses (published in the last five years) was randomly selected, and RCTs that composed them were collected and, only when the respective authors returned congruent results, their application was initiated.

### 2.5. Statistical Analysis

Meta-analysis was performed using STATA^®^, version 13.1 (StataCorp, College Station, TX, USA).

Independent meta-analysis was conducted for the outcomes most investigated in included studies (FFM and FM) from mean and respective standard deviation quantified before the beginning of intervention and after supplementation (if the study was designed to measure the outcomes at various moments after supplementation, the value measured on the day that supplementation ended was used). All studies reported that there was no difference between intervention and comparative groups at baseline. Effect sizes between treatment and comparative groups were calculated based on weighted mean differences (WMD) and respective 95% confidence interval. No imputation measures for missing data were applied.

Heterogeneity was deemed significant if *p*-value determined by the chi-squared test was lower than 0.1 or I-square test (I²) greater than or equal to 50% or visual inspection of the forest plot showed studies with discrepant effect sizes and confidence intervals, comparing to other studies. Meta-analysis was performed using a fixed-effects model (using the Mantel-Haenszel method) when heterogeneity was not significant and a random-effects model (using the DerSimonian-Laird method) was used when heterogeneity was significant.

Subgroup analyses were performed for outcomes in relation to the type of whey protein evaluated, the protein percentage and the level of physical activity. Sensitivity analysis to individual studies (leave-one-out method) was performed to verify the influence of each study on global WMD (made only for pooled analysis for each outcome). Publication bias was not evaluated since the minimum number of studies needed was not reached (*n* = 10) [30].

*p*-value < 0.05 was considered statistically significant.

## 3. Results

### 3.1. Searches, Screening, and Quality Analysis

Our search identified 5250 results in addition to another 49 that were collected from the bibliographic references of these studies, totaling 5299 articles. At the end of the screening, 10 studies met all the eligibility criteria to compose the systematic review [31,32,33,34,35,36,37,38,39,40]. However, in two cases [39,40], data collection was incomplete even after successive contacts with the authors, which resulted in eight studies included in the meta-analysis. The PRISMA flowchart and the complete search strategy are available in Figure 1 and Appendix A, respectively.

In the parallel search, 640 registration protocols for RCT were identified, but after analysis, none met the criteria necessary for their inclusion in this research. The complete detailing of quantitative results found that screening and selection of both searches can be found in Appendix A.

The schematic representation of the overall risk of bias analysis result is shown in Figure 2, and results individualized for each domain are available in Appendix A. Considering that only one study with low risk of bias was evaluated [32], it could be inferred that, in general, RCTs included in this meta-analysis presented low methodological quality.

### 3.2. Methodological Characterization

All studies included in this research were randomized, double-blind, placebo-controlled, parallel-arm trials published from 2001 and, except for two studies (Australia [32] and Finland [34]), which were conducted in Anglo-Saxon countries. A descriptive summary of intervention patterns and population of selected RCTs is described in Table 2, and the FFM and FM values at the initial and final moments for both groups, as well as individual delta values, are available in Table 3.

A total of 288 individuals were included in the sample, 150 of them were allocated in the WP intervention and 136 in comparative groups. The sample size ranged from 12 [32] to 45 [38] participants, and only one study showed disproportionate losses between groups and at the end of this study [31], the placebo arm contained twice as many individuals compared to the intervention group.

All studies evaluated in the meta-analysis had experimental design performed only in men [31,32,33,34,35,36,37,38]. Two RCTs had women in their sample [39,40], however, these studies were the same that could be included only in the qualitative evaluation due to data incompleteness. In general, the age group of participants was from 18 to 54.2 years; however, mean value and standard deviation per randomized group could not be calculated since not all RCTs showed this information in the text.

Studies were all performed in healthy adults, and all used some method to certify the candidates’ health status: 50% of studies used questionnaire or interview during screening [31,32,33,36,37,39] and the others also used physical or laboratory tests [34,35,38,40]. In all cases, none of the randomized individuals presented life habits, diagnoses, or pathologies that influenced their cardiometabolic health and/or body composition.

The previous level of physical activity of participants varied among studies, however, all included physically active individuals: three with sporadic practitioners [34,38,40], five with regular practitioners [31,32,33,35,36], and two with athletes [37,39]. The cutoff point adopted by RCTs for frequency and regularity of practice of physical activity varied from two to three times a week for the first two categories and participation in professional championships was used as a criterion for the latter.

### 3.3. Nutritional Intervention

Considering RCTs included in the qualitative analysis, a balance regarding the type of whey protein supplement was observed: 50% used WPC [33,35,39,40] while the others used WPI [31,32,36,38]. For the meta-analysis, as shown in Table 2, one study used a second randomized group with WPH administration, which allowed its double inclusion in the quantitative investigation [35]; however, this was the only RCT among those selected who made use of this variant and this prevented the evaluation in meta-analysis for hydrolyzed whey protein.

In general, the daily dose of WP intervention in selected RCTs varied from 0.24 [37] to 1.28 g·kg^−1^ [32], however, two studies [31,32] presented doses higher than 1.0 g·kg^−1^ in comparison with the others, whose values were equal to or lower than 0.27 g·kg^−1^. Nevertheless, all RCT implemented the same dose for both placebo and intervention groups during the entire experimental protocol. The total nutritional intervention duration, in mean and standard deviation, was 46.75 ± 12.46 days of supplementation. The percentage protein composition of each type of WP for RCTs included in the meta-analysis was also evaluated, however, in one case, it was not possible to obtain information from the corresponding author [31]. From extracted data, two studies evaluated WP with protein percentage in the range from 51% to 80% [34,35], three in the range from 81% to 95% [36,37,38] and two [32,33] used products with percentage in the range from 96% to 100%.

Although three studies [32,36,37] did not inform the specific type, the control arms of all 10 RCT used carbohydrate placebos of commercial origin, namely: maltodextrin [31,33,34,38,39,40] and dextrose [35]. In all cases, as part of the inclusion criteria, studies were in the isocaloric condition compared to their respective intervention groups.

In a methodological comparison of studies included in the meta-analysis, the standardization of the supplementary intervention protocol was heterogeneous: one study reported that the supplementation was administered in four portions throughout the day, however, it did not specify the exact times [31]. another reported that supplementation was administered in three portions throughout the day: early in the morning, in the afternoon, and in the evening [32]; two reported administration twice a day before and after training [33,35]; two studies adopted daily dose with intake restriction before breakfast or after training [36,37]; and, finally, two other studies described that supplementation occurred after training [34,38].

In all cases, no significant differences were found among groups regarding adherence of participants to the supplementary intervention protocol, either protein or carbohydrate.

Although the form of presentation of results differed, protocols for evaluating diet were similar among studies: 10 RCT reported that participants were instructed to maintain their routine food intake combined with the completion of a registration form of foods consumed during the research.

Despite this methodological equivalence, only two studies included in the meta-analysis explained in the text the restriction of concomitant intake of other supplements in addition to WP prescribed during intervention [36,37]. Moreover, another study restricted the consumption of caffeine and derivatives during the period of protocol execution in order to avoid neutralization of the effects of creatine on the metabolism of participants [31]. After analysis, however, we concluded that the food registry results presented by studies did not differ among groups that had their data extracted for this review.

### 3.4. Physical Intervention

All studies included in the meta-analysis implemented effort routines, analyzed and/or supervised by professionals, with control points for exercise compliance. Exercise intensities used were medium to high according to the number of repetition maximum (RM) at baseline moment. Additionally, RCT was performed in triathletes and, in this case [37], the routine of usual training series at three intensities according to ventilatory threshold (VT2) was used as physical intervention protocol. A descriptive summary of aspects related to trainability implemented in all studies is available in Table 4.

The total duration of training protocols was 64.5 ± 15.3 days, and the number of training sessions per week was 3.17 ± 1.0 trainings. Of these values, expressed as mean ± standard deviation of studies included in the meta-analysis, data referring to training performed during the adaptation period—which was implemented in all studies—are already discounted.

All physical interventions started simultaneously with the first day of food supplementation and no study reported significant differences regarding adherence of participants. These results allow us inferring the existence of homogeneity regarding the implementation of physical activity in studies included in this review.

Moreover, both RCTs implemented training routines only qualitatively, for the entire body, similar to each other [39,40], being composed of resistance exercise combined with the practice of flexible models of nonlinear agility. Only one of these trials reports restriction to the practice of physical activity outside the training protocol [39], which is the non-performance of resistance exercise in the 48 h prior to training days.

### 3.5. Effect of Whey Protein on Body Composition

Overall analysis showed a statistically non-significant increase of FFM (WMD = 0.26; 95% CI = −0.32, 0.83; *p* = 0.381) (Figure 3a). Considering methodological issues capable of affecting the results obtained, a sensitivity analysis of individual studies (leave-one-out method) was performed (Appendix A). The exclusion of studies (one by one) did not modify the significance of results. Stratified analyses according to the type of whey protein provided, percentage of whey protein and level of physical activity did not show statistically significant results for FFM (Figure 4a, Figure 5a and Figure 6a).

Conversely, overall analysis showed a statistically significant decrease of FM (WMD = −0.96; 95% CI = −1.37, −0.55; *p* <0.001) (Figure 3b). A sensitivity analysis of individual studies (leave-one-out method) was also performed (Appendix A). The exclusion of studies (one by one) did not modify the significance of results.

Stratified analysis according to type of intervention was performed, showing a statistically significant decrease of FM with the use of whey protein concentrate (WMD = −0.63; 95% CI = −1.19, −0.06, *p* = 0.030) (Figure 4b). Stratified analysis according to the percentage of WP showed a statistically significant decrease of FM when protein percentage varied from 51% to 80% (WMD = −1.53; 95% CI = −2.13, −0.93, *p* <0.001) (Figure 5b).

Lastly, stratified analysis according to level of physical activity showed a statistically significant decrease of FM for regular exercisers (WMD = −0.95; 95% CI = −1.70, −0.19, *p* = 0.014) (Figure 6b).

### 3.6. Adverse Effects

Although our PROSPERO protocol established collecting data regarding side effects and/or adverse effects of WP ingestion, only three studies analyzed these data, and no deleterious effects were observed in these cases.

Even though evaluating losses of participants during the performance of each RCT investigated in the meta-analysis, it was identified that 65.63% of 64 reports occurred during the supplementation stage. In cases where the reason for withdrawal was informed (71.88%), only 2.17% (*n* = 2) were related to the food protocol. Therefore, considering the limits of this analysis, it was possible to observe the non-significant occurrence of adverse effects in the study population.

## 4. Discussion

After extensive scanning of significant scientific bases, it was concluded that, to our knowledge, this is the first meta-analysis that sought to investigate the efficiency of WP supplementation among the main variants consumed of this product, comparing it to isocaloric placebos in physically active adults. Considering that hypertrophy is the main objective of the majority of WP consumers and its relative high cost in comparison to other non-protein supplementation strategies [41], this metanalytic approach adopted here acquires distinct relevance for the clinical sport practice.

Our results show that, in this situation, there is no positive effect for FFM gain regardless of type of WP used, its protein percentage, or level of physical activity. On the other hand, the reduction in FM is significant, occurring only for WPC and for regular physical activity practitioners. In addition, it was found that this reduction presents a trend inversely proportional to protein content, being verified only in reduced values. Except for one analysis, the heterogeneity among studies was not significant.

### 4.1. Methodological Approach of the Problem

Some types of protein have the potential to influence the entire protein metabolism by modulating muscle development through physical activity. Its intrinsic characteristics, such as absorption rate, amino acid profile, hormonal response, and antioxidant potential, have a direct effect on muscle gain [42,43]. Thus, the daily dose ingested becomes a determining factor for hypertrophy [16].

In order to reach recommendation levels, most athletes regularly use WP as a food routine component [6,7,8,9,10], and frequently this consumption, without a correct nutritional accompaniment, can occur in imprudent doses [9,15,17]. In our analysis, the establishment of a single standard recommendation for WP users has two direct consequences: possible occurrence of health damage as a result of overdose (a situation not reported by any of RCTs included in this review) and lack of adequacy to the individual protein demand. An orientation established only in gross weight WP values does not consider the physical characteristics of each athlete [15,16,17]. For this reason, aiming at better adequacy, some reviews stipulate daily protein intake values based on the individual’s body composition. To maximize skeletal mass gains and protein synthesis, values of ~0.4 g·kg^−1^ are indicated for young adults and ~0.6 g·kg^−1^ for older adults per meal, considering minimum of four throughout the day in order to reach daily consumption of approximately 1.6 g·kg^−1^ [43,44]. Another example is the indication for those who are in caloric restriction, which ranges from ~1.8 to 2.7 g·kg^−1^ or even from ~2.3 to 3.1 g·kg^−1^ of FFM [38]. The studies included in the meta-analysis implemented WP doses between 0.24 and 1.28 g·kg^−1^.

Although we chose to present the dose implemented in grams per kilogram of mass of the individual in each study (aiming to show more representative data of the individual daily consumption), not all selected RCT observed in the same way the food intake of participants, especially regarding the inclusion or not of protein supplementation values in the final calculation. In addition, although it is possible to infer that all adopted restriction on the consumption of other supplements, only two studies explained this result in the text [36,37]. Consequently, it became unviable to reasonably assess the adequacy of the administered WP dose compared to values recommended for total proteins in literature for this population. However, we emphasize that the food consumption analysis of each study was the same used for all randomized groups and that, according to our evaluation, the intergroup results of none of the studies showed relevant differences in order to significantly influence participants’ body composition.

In addition, the dose (in grams) and the total time of supplementary nutritional intervention (in days of consumption) were also similar to studies found in other reviews [9,44]. Although it does not allow making definitive conclusions regarding protein adequacy for physical activity practitioners, this result similar to literature allows inferring the linearity among intervention protocols that have been implemented by the scientific community. However, we consider that future RCTs should standardize the form of food consumption analysis of participants regarding presenting, preferably in grams per kilogram of mass, both the supplemented ingestion values and also the total amount disregarding the established by their intervention protocols.

Considering the importance of aspects related to protein supplementation, the nutrient timing is also one of the factors that have been gaining the focus of debate in the scientific community [45,46]. In our review, this was one of the domains of analysis that varied most among studies, both in terms of portioning and distribution, being one of the main reasons that made us choose the random-effects model for the meta-analysis. The consensus regarding the real efficiency of each ingestion distribution model is still far from being reached, however, some qualitative reviews have indicated that the consumption of moderate doses of protein supplements in the pre-sleep moment, combined with physical activity throughout the day, favors the increase in muscle protein synthesis [45,47,48]. Although it is necessary to consider the inherent difficulties of conducting randomized clinical trials adopting multiple protein portioning distributions, our investigation points to the need for greater standardization of future supplementation protocols in order to obtain further conclusions about it.

According to the values found, we chose to establish cutoff points for the percentage of WP supplemented in each study: 51%–80%, 81%–95%, and 96%–100%. Although limited due to the inherent heterogeneity among the studies, the option for this fractioning occurred after verifying a worrying data about the WP production chain: disregarding the study in which it was not possible to obtain these data [31], when comparing values identified to cutoff points available in literature (already reported in the introduction of this article), it was found that the protein content evaluation, although all RCTs have performed it, it was not taken into account with one of the criteria for the distinction between WPC, WPH, and WPI in studies included in this review.

To exclude this bias, as previously described in Section 2.1, the protein typology was confirmed using both classification criteria through descriptive information contained in the text of each study comparing these findings to the values on the product label on the website of each manufacturer. This allowed a more broad stratified analysis of data, also focused on bioavailability, but on the other hand, introduced a limiting factor in the research. This dichotomy shows that WP classification according to its typology is more usually related to the manufacturing process than to its final protein content. The divergence in the legislation that regulates the trade of food supplements in each country is notorious. Instead of having specific legal mechanisms to ensure their regulation, most countries include food supplements, such as WP, in subsets of existing legislation for biological, herbal or food compounds [49]. In addition to hindering inspection, this delays progress in establishing uniform standards for the proper identification and classification of these products in the global food industry [6,7,8,49].

A system of specification of WP variants according to both criteria brings the advantage of being more adequate to the reality of consumption of this supplement by physical activity practitioners, while the use of the manufacturing criterion only, can lead to the marketing of products with levels below adequate for hypertrophy achievement. The use of legislation aimed at standardizing the marketing of WP by adopting its classification by both protein percentage and fractionation method is therefore essential. Until then, a legal precedent has been set for WP manufacturers to include their products in categories that are not consistent with recommendations in scientific literature. Consequently, new systematic reviews on the subject should also investigate this aspect in order to categorize WP products within implemented interventions.

Regarding the nutritional aspects of the methodological approach, it became essential to evaluate the choice of supplementation protocol of control groups adopted by studies included in this review. As previously mentioned, the selected RCTs adopted commercial carbohydrate placebos of two types: dextrose and maltodextrin. Since all of them were in an isocaloric condition compared to the protein group and no study reported significant differences in participants’ adherence to food intervention, it could be conclude that the placebos used were virtually identical.

In addition to aspects related to food routine, physical activity, and parameters related to trainability are also factors that play an essential role in muscle gain [50]. As part of the criteria adopted in the screening of this review, only studies involving physically active individuals were selected. This is because the metabolic physiology of the energy expenditure is different in sedentary individuals and, in addition to being regulated by genetic factors, its modulation is gradual according to the level of adaptation to the physical exercise intensity [50,51].

Natural mechanisms of biological compensation such as increased energy expenditure at rest by thermoregulation increased hunger in fasting and intestinal growth modulation are usually expressed in individuals in transition from sedentary to physically active condition, and also from this to physical rigor required by more exhaustive and frequent training routines [51,52,53,54]. Such mechanisms hinder sudden metabolic changes during transition and adaptation stages, so that the general metabolism always tends to be focused on ensuring the maintenance of the homeostatic energy state before routine change, whether of dietary origin or muscle work [51,55,56]. Therefore, there is need to implement a period of adaptation to training in studies that have physical activity as protocol and, although it was a methodological similarity casually produced by screening, in our review, this step was present in all selected RCTs. Given its importance for the energy metabolism regulation, further reviews should evaluate the existence of this adaptive phase, defined as part of the inclusion criteria.

Nevertheless, the quantification of the losses of RCT participants included in the meta-analysis showed that, in general, 34.67% of them occurred in this adaptive period and that of these, 31.05% were directly related to physical activity. Since all individuals were physically active before participation the study and all participants had similar previous training routines, these results point to the need to introduce evaluation criteria for this adaptive period in future clinical trials in order to investigate their effectiveness.

Since the level of physical activity has a strong influence on the adaptive and/or physiological resistance to the increase in individual energy expenditure before new stimuli [57,58], we chose to perform the analysis in three categories regarding this factor: resistance and/or recreational, regular, and athletic strength exercises. This classification took into consideration aspects related of training in each stage of this classification. Factors such as number of sets and replicates per set, execution speed, exercise ordination, and interval between executions, for example, differ according to the individual’s training stage so that the adequate choice is determining to increase the adaptive capacity of athletes [50,57].

Seeking to apply this investigation to the daily life of most physically active individuals, the implementation of resistance and/or strength exercise was established as one of the inclusion criteria adopted in our study. An investigative assessment of intervention protocols, as previously pointed out (Table 4), reveals a positive relationship between trainability characteristics and volume and intensity, which are, in turn, the main factors related to training efficiency [50,57]. Moreover, the average total duration of the training routine implemented in studies was also similar to values found in other reviews [9,44,57]. These results, combined with the fact that there were no statistically significant differences among groups regarding adherence to the training protocol, it could be concluded that there is homogeneity for this field of analysis.

However, an essential methodological failure was observed in the practice of physical exercise: while Naclerio et al. [37] used the usual training routine of participants as a physical activity protocol (since they were professional athletes), and two other studies did not mention any information about it [31,35], none other RCT established this restriction. In addition, of the studies showing physical activities outside the protocol, only that of Cribb et al. [32] investigated data using a daily reminder filled by participants. Although the analysis of this record did not indicate significant differences (excluding data from individuals with incorrect and/or insufficient filling), it was considered that this gap introduced substantial bias of non-differential information, which is the main limitation of this meta-analysis.

In this systematic review, only clinical trials whose randomized individuals were adults at the time of acceptance were selected. However, some studies indicate possible metabolic differences even within this cutoff point (18–60), especially after 40 years of age [59,60]. However, of studies included in this review, only one covered this age group [37] and, therefore, a more in-depth analysis of the influence of age on the effects of WP would inevitably be limited. In addition, sensitivity analysis showed that the overall result does not change with the absence of data from this and any other clinical trial.

Finally, after assessing the risk of bias, it was concluded that RCTs were of relatively low methodological quality. However, some factors lead us to make an important observation regarding this result. Perhaps as a consequence of restrictive choices in narrative selectivity or the limit of characters imposed by several publishers, almost all the authors did not report in the text information which, although indirectly perceptible in some cases, would contribute to a more positive analysis of the risk of bias. A clear example is the omission of the restriction of parallel consumption of supplements other than those provided by the intervention protocol.

Considering the amplitude of the interpretative limits of the ROB 2.0 tool [28], and even though it is possible to infer some assumptions through the analysis of other descriptive elements, the mere deduction of the occurrence of a methodological process does not provide sufficient support in this case to reach a definitive conclusion in this regard. Thus, it is emphasized that much of this result is due to the report quality, and not necessarily to the actual occurrence of methodological failures in conducting the studies included in this review.

### 4.2. Effect of Whey Protein on Body Composition

An analysis of the individual results of each RCT (summarily presented in Table 3) reveals a stable increasing trend in FFM gain both for control and supplemented groups comparing their respective final and initial moments. However, the delta value was not significant (*p* > 0.05) comparing groups with each other. This result was maintained in this global effect meta-analysis, as shown in Figure 3a. As in each study, individuals were submitted to the same training conditions and isocaloric conditions compared the opposite group, this result demonstrates that the protein isolate derived from WP was not sufficient to provide resistance exercise practitioners a more considerable increase in FFM when compared to placebos. Therefore, it could be inferred that in this situation, the positive effect on body composition is more related to the efficient practice of physical activity than to protein supplementation.

In the individual FM evaluation in each RCT (Table 3), a persistent gain tendency is verifiable for placebo groups while the opposite was observed for protein groups. In addition, studies included in the meta-analysis reported significant statistical difference (*p* < 0.05) for FM loss in WP groups compared to placebo-controlled groups. Furthermore, a visual inspection of the graph of this global meta-analysis (Figure 3b) reveals that this tendency in FM gain in the comparative groups is evident so that all diagrams point to WP supplementation benefits.

Comparing these findings with the effects on body composition described in similar meta-analyses that evaluated WP supplementation allied to resistance exercise, wide divergence of results was observed.

Schoenfeld et al. [61], in a sample of 525 individuals in 23 studies, analyzed the influence of the daily protein intake distribution, with or without supplementation, comparing to controls with or without protein in their composition. There was no categorization for type of WP, and the caloric equivalence did not constitute the eligibility criteria. In this scenario, there were no significant differences in the global analysis either for hypertrophy (*p* = 0.18) or for FFM in the model reduced to covariates (*p* = 0.27).

Naclerio and Larumbe [62], in a meta-analysis that used calorie equivalent placebos compared to the protein group, but did not stratify results by type of WP, did not find significant statistical differences for body composition (FFM or FM) among studies that supplemented WP with no associations, while significant effect was verified for combined interventions (*n* = 4, g = 0.468, 95% CI 0.003–0.934). In this sample composed of 192 participants in 9 studies, according to the adopted inclusion criteria, control groups could or could not be supplemented with compounds of protein origin, except for WP.

Nissen and Sharp [63] adopted the presence of WP intervention as inclusion criteria, which may be associated, but only with non-anabolic substances. Placebos used could or could not be caloric and, moreover, there was also no analysis for type of WP. After screening, 48 designs in 40 clinical trials were selected, whose participants could or could not be previously trained with resistance exercise. Of these studies, four had physical activity as part of the intervention protocol and, in this sample, both LM and FM were not affected by protein supplementation (IC: 0.07–0.31%, *p* = 0.31 and CI: 0.87–0.51%, *p* = 0.66, respectively).

Miller et al. [64] performed a stratified verification of results by type of WP (concentrated and isolated) and found no significant differences (*p* > 0.05) in the body composition of participants. Nevertheless, placebos used were not paired for calories. In their global findings, a non-significant increase in LM in the order of 0.83 kg (95% CI, −0.36–2.03) was observed in a sample of seven studies.

Morton et al. [37] in their metanalytical review (consisting of 1863 participants in 49 studies) in which there was no calorie equalization of comparative groups as well as no analysis for type of WP, protein supplementation promoted statistically significant changes in FFM (MD: 0.30 kg (0.09, 0.52), *p* = 0.007) and FM (MD: −0.41 kg (−0.70, −0.13), *p* = 0.005).

On the other hand, the most recent meta-analysis performed on this subject [65], still in the publication process, in an aggregate of 21 RCT in ~13.1 weeks of resistance training, detected significant effect for LM (*p* = 0.01) with an increase of 0.46 kg, and significant reduction of FM 0.62 kg (*p* = 0.004) in individuals supplemented with WP compared with placebos. An analysis of studies that composed this specific review reveals that only 33.33% (*n* = 7) implemented food protocols in order to calorically equalize branches of each intervention. Of these seven clinical trials, five were also included in our meta-analysis [31,35,36,37,38], with the others were Taylor et al. [39] and Volek et al. [40], who composed only our qualitative review for reasons already reported in this article. In addition to positively strengthen the screening performed here, it also reinforces the hypothesis verified in this meta-analysis that isolated and/or concentrated WP may not have a significant effect on FFM gain in comparison to non-protein isocaloric interventions.

### 4.3. Limitations, Strengths, and Quality of Evidence

This systematic review and meta-analysis have several factors that limit its direct application in clinical sport practice. RCTs that met all eligibility criteria had systematic failures in the reporting of information so that they were evaluated, in general, with low methodological quality. Moreover, intervention protocols were different from each other in terms of effecting, restricting, and evaluating or not food consumption and the practice of physical activity other than those proposed in this research. Although the protein composition of WP was evaluated, in one case [31], it was not possible to obtain information from the corresponding author. Furthermore, it was considered that the confirmation analysis of WP typology, even in pairs and blindly, is inherently exposed to an evaluator’s bias.

Regarding the study sample, two points hinder the juxtaposition of our results. All essays included in the meta-analysis were performed only in men, which inevitably restricts our findings to only a portion of the physically active population. Another factor is that, considering the country of origin of each RCT (Table 1), it was observed that all are among the 15 best evaluated in the global human development index [66]. Considering that a substantial portion of recruited individuals were native, this result exposes our review to a significant geosocial limitation since, in addition to genetic components and environmental aspects, factors such as family income, cultural diversity and social habits play an essential role in the diet quality and diversification and have a potential influence on the metabolic profile of the population [67,68].

On the other hand, our review and meta-analysis also has several strengths that reinforce the power of its findings: the amplitude and magnitude of the effects of WP supplementation on physical activity, both for FFM and FM was very similar among studies and, moreover, sensitivity analysis showed that the overall result is not altered by the absence of any RCTs, which therefore demonstrates strong consistency. The methods for body composition measurement were selected only by the gold standard, were in a condition of similarity among studies, and, moreover, were blindly performed in all clinical trials. The intervention protocols showed congruence and similarity for the primary analysis domains of proven direct influence on body composition. Placebo intervention of comparative groups was virtually identical among studies; and, finally, in almost all analyses, heterogeneity among RCTs can be considered low, very low or, in some cases, potentially null with I^2^ values equivalent to 0%.

Considering the balance between consistency, accuracy, and amplitude of results of each RCT, the implications of identified bias, and limitations of this review, we chose to apply the GRADE method in order to enable the better understanding of results found in this metanalytic study. Given that the publication bias was not statistically investigated due to the number of studies included in the quantitative analysis, scaling of domains related to similarities of estimates, overlap of confidence intervals, results of heterogeneity tests was performed, considering the fact that this systematic search also included the search for registration protocols for clinical trials in three scientific bases and that, of the 266 results assessed by eligibility criteria (total of 640), none was finalized and not published. This allowed us inferring a low risk of publication bias for all investigated outcomes.

After schematizing the evidence profile of studies included in this review, results were compiled in the form of a synthesis of conclusions that are available in Table 5. Based on results, the following could be concluded:Low power of evidence, in which WP does not significantly change FFM when compared to other non-protein isocaloric interventions and that concentrated and isolated WP, even with different protein levels, are virtually identical regarding the amplitude of this effect;Moderate power of evidence, which after adapted to resistance exercise, the non-occurrence of FFM gain regarding WP supplementation, both in its concentrated and isolated form, does not depend on the level of physical activity;High power of evidence, in which WP supplementation is favorable to the loss of FM and this reduction is observed only for WPC;Moderate power of evidence, in which the depletive effect of FM occurs only in WPs of low protein content (51%–80%); and this result is verified only in regular practitioners of resisting and/or strength activity;Very low power of evidence, in which there is an inversely proportional trend between protein WP content and FM reduction.

### 4.4. Future Recommendations

Evaluating the scope and diversity regarding multiple factors related to the use of WP, as well as the exchange of its importance in areas that go beyond sports such as public and preventive health, further new randomized clinical trials and meta-analyses are necessary for a better understanding of the effects of WP supplementation on relevant outcomes for human health, such as body composition. In order to contribute to the advancement of knowledge in this area, some recommendations discussed in our review are highlighted below that in our analysis, may support the methodological standardization of further studies.

After screening using eligibility criteria, there was only one randomized branch of one study that used WPH as intervention protocol. Although we recognize that, among all types, WPH has the highest financial cost. Our analysis is that studies with this supplementary variant are vital to advance knowledge in this area. Although all nutritional intervention protocols showed the analysis of participants’ food intake outside the study, these data were not assessed in the same way by RCTs. Thus, if a standardized laboratory diet developed by a qualified nutrition professional is not feasible, we believe that future studies present, preferably in grams per kilogram, both supplemented intake and total quantity values, disregarding values established in their intervention protocols. Furthermore, since some studies collected data from other feedings, in only three moments and even though food registration protocols already validated for various populations are abundant in scientific literature, we strongly recommend that this evaluation should be performed at least weekly during the research.

Even after decades of studies (the oldest WP supplementation clinical trial identified dates back to 1983), there are still some gaps. In our discussion, a possible influence of the moment of supplementation ingestion on muscle gain was identified and, in this sense, we consider that RCTs should adopt multi-time distribution so that new and more robust meta-analyses can be performed.

We also identified that placebos should be more standardized. Given that the effect of WP compared to controls is already well established, we recommend that new studies, including meta-analysis, also adopt caloric equivalence of placebo compared to groups supplemented with WP.

As previously discussed (Section 4.1), it is essential to use combined legislation to standardize the marketing of WP by adopting its classification both by the fractionation criterion in the production chain and also by its final protein content. Meanwhile, we consider it valid that upcoming reviews adopt an analysis, in pairs and blind, of the WP type supplemented in the RCTs contrasting, whenever possible, the information described in the text with the centesimal composition available in the manufacturer’s website. Besides, we also consider it prudent for the next clinical trials to perform analyses of the WP composition by independent laboratories in order to corroborate the information of the product label to be supplemented.

Regarding physical activity, we believe that next RCTs in the area should adopt more selective analysis criteria in their protocols regarding the performance of parallel physical activity. Some studies included in this review, despite implementing this restriction, allowed the practice of recreational activities. In our evaluation, this concept can often be broadly interpreted, especially considering that, at the time of execution, participants are not supervised by the researcher’s team. In an attempt to reduce this bias, we recommend the choice of two most appropriate situations according to results obtained: (a) total restriction of non-labor physical activity (more indicated but less feasible methodologically); and (b) restriction of physical activity of the same typology implemented in the study, adopting as part of the protocol in this case, the daily completion of a detailed form with information on exercise intensity and volume in order to allow the statistical evaluation of data among groups.

In our review, substantial losses of participants during the adaptation training period were found, therefore, future studies should adopt criteria to assess the effectiveness of this adaptive step in order to ensure that all randomized individuals are comparable in terms of physical capacity to perform the exercise protocol. In addition, we believe that further reviews should adopt this phase as an inclusion screening criterion.

Almost all RCTs included here were assessed with low methodological quality due to the risk of bias and, as seen in Section 4.1, it was considered that a large part of this result was an inevitable consequence of the quality of the study reports. As a result, we strongly recommend that the authors of clinical trials should adopt less rigid criteria in the selectivity of information to be published in order to enable not only a better understanding but also a better judgment of the possible risks of bias.

In our literature searches, it was identified that of the 114 RCTs included in our final screening and the 86 systematic reviews with meta-analysis, only 20.17% and 37.21% were included in publicly accessible digital platforms. The publication of a study not previously registered, in addition to increasing the chances of the occurrence of biases and hindering the proper understanding of its execution by the reader, exposes the scientific community to the risk of redundancy, so that, without this record, the likelihood of more than one research group performing similar studies is very high. The publication of a study protocol in advance in networks such as PROSPERO for reviews and Clinical Trials for randomized studies (both global and free) may significantly contribute to reducing the use of material and human resources in science.

Moreover, the need to extend the domains of analysis to other variables related to physical activity was also identified. We believe that investigating and reporting sociodemographic information on the population recruited in the next RCTs would significantly contribute to expanding the scientific knowledge in the sports area.

Finally, we would like to emphasize the latent need for RCTs to be carried out with the female public in the sports field. Since 2011, only in the African continent, for example, the physically active population between men and women was already comparable (83.8% and 75.7%, respectively) according to criteria defined by the World Health Organization [69]. It is a consensus that, although growing, the number of studies focused on the sports field including women is unfortunately far from achieving a minimally realistic representation of the physical exercise population. We consider that this is one of the leading and most urgent challenges to be overcome in this area by the scientific community.

## 5. Conclusions

This systematic review identified 10 RCTs in which WP supplementation was applied compared to isocaloric placebos in resistance training protocols. No significant effects were found to FFM gain in any of the scenarios investigated, and a moderate FM loss was significant relative to WPC supplementation by regular physical activity practitioners. Despite its consistency and low statistical heterogeneity, these meta-analysis results are limited to eight studies, so that further RCTs based on this isocaloric approach are needed in order to provide more detailed evidence, especially regarding FM gain. These findings also should be interpreted preliminarily due to the several limiting factors (intervention methodological variability, sample restriction, high risk of bias, etc.) being discouraged, therefore, their direct application in clinical practice. Besides, it is suggested to investigate the extent effect of more variants of the WP, also focusing on the age, gender, social conditions, ethnic groups and lifestyle of the physically active population.

## Figures and Tables

**Figure 1 nutrients-11-02047-f001:**
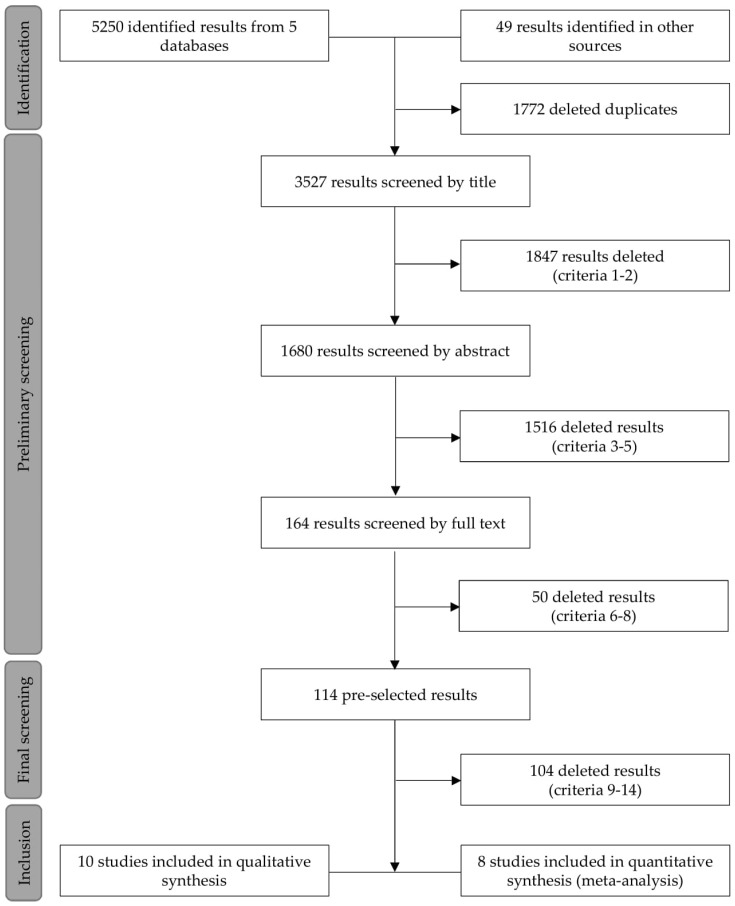
Preferred Reporting Items for Systematic Review and Meta-analysis (PRISMA) information flowchart for the selection and screening of primary studies. (1) Randomized clinical trial; (2) English language; (3) healthy individuals; (4) adults between 18 and 60 years old; (5) non-sedentary; (6) intervention of isolated, concentrated, and/or hydrolyzed whey protein; (7) intervention of physical activity of strength and/or resistance; (8) total interchange of participants between the groups for cross-over studies and sufficient time between the rounds of the experiment; (9) analysis of body composition between lean, fat-free and fat mass; (10) analysis of body composition by gold standard methods; (11) whey protein intervention, without associations, compared to placebos; (12) comparative group equated to whey protein intervention for calorie; (13) absence of variables parallel to listed interventions; and (14) non-inclusion of multiple publications.

**Figure 2 nutrients-11-02047-f002:**
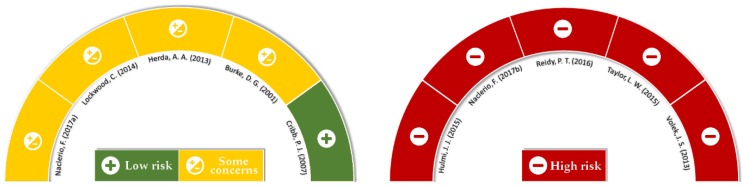
Analysis of global risk of bias (RoB 2.0).

**Figure 3 nutrients-11-02047-f003:**
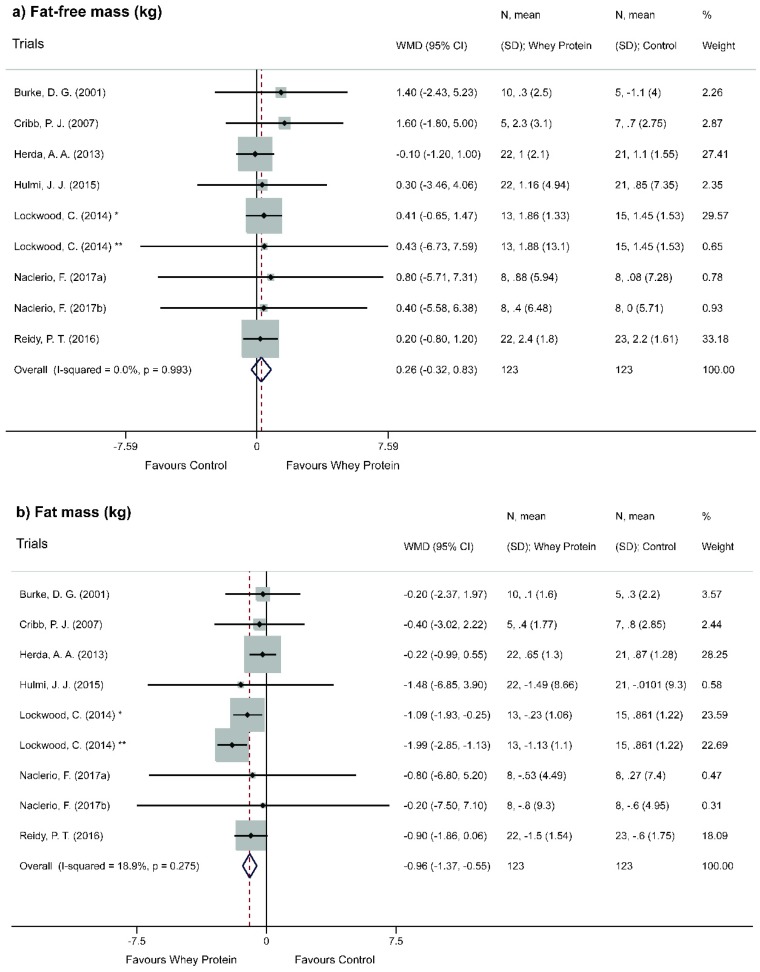
Meta-analysis of the overall effect of Whey Protein supplementation on body composition. (**a**) Fat-free mass. (**b**) Fat mass. * Whey protein concentrate. ** Whey protein hydrolyzed.

**Figure 4 nutrients-11-02047-f004:**
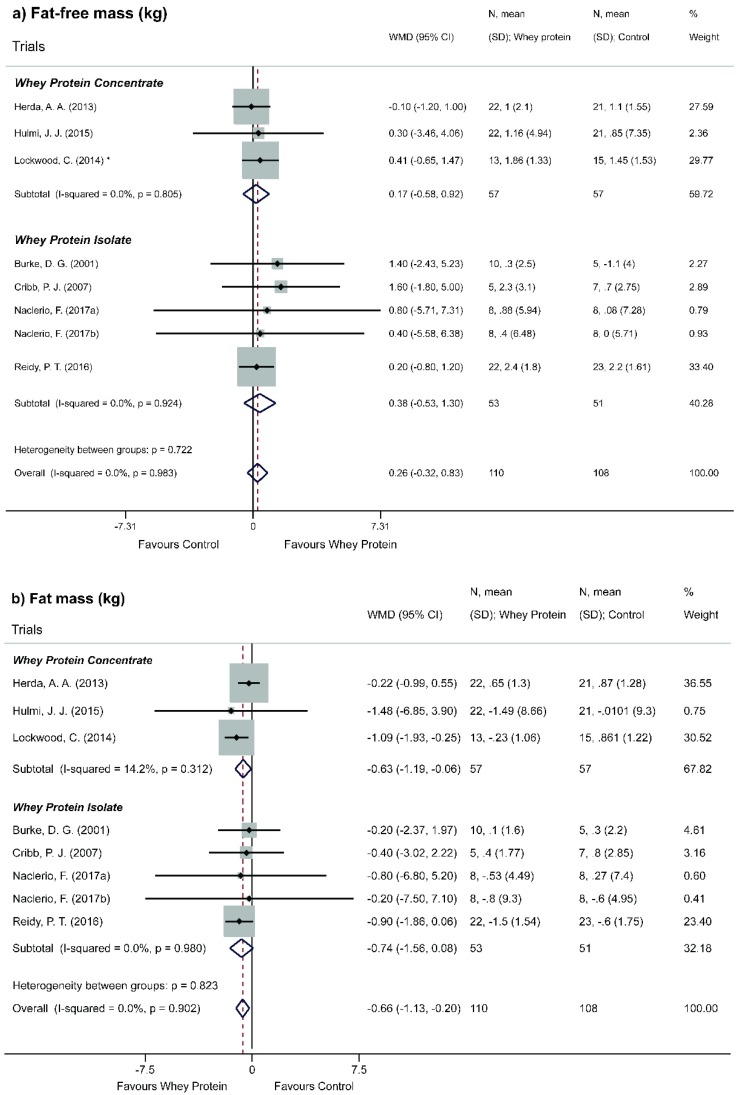
Meta-analysis of the effect of Whey Protein in subgroups for typology. (**a**) Fat-free mass. (**b**) Fat mass. * Whey protein concentrate. One of the samples of Lockwood et al. [35] was omitted from this analysis because it was the only study in which individuals were supplemented with hydrolyzed whey protein.

**Figure 5 nutrients-11-02047-f005:**
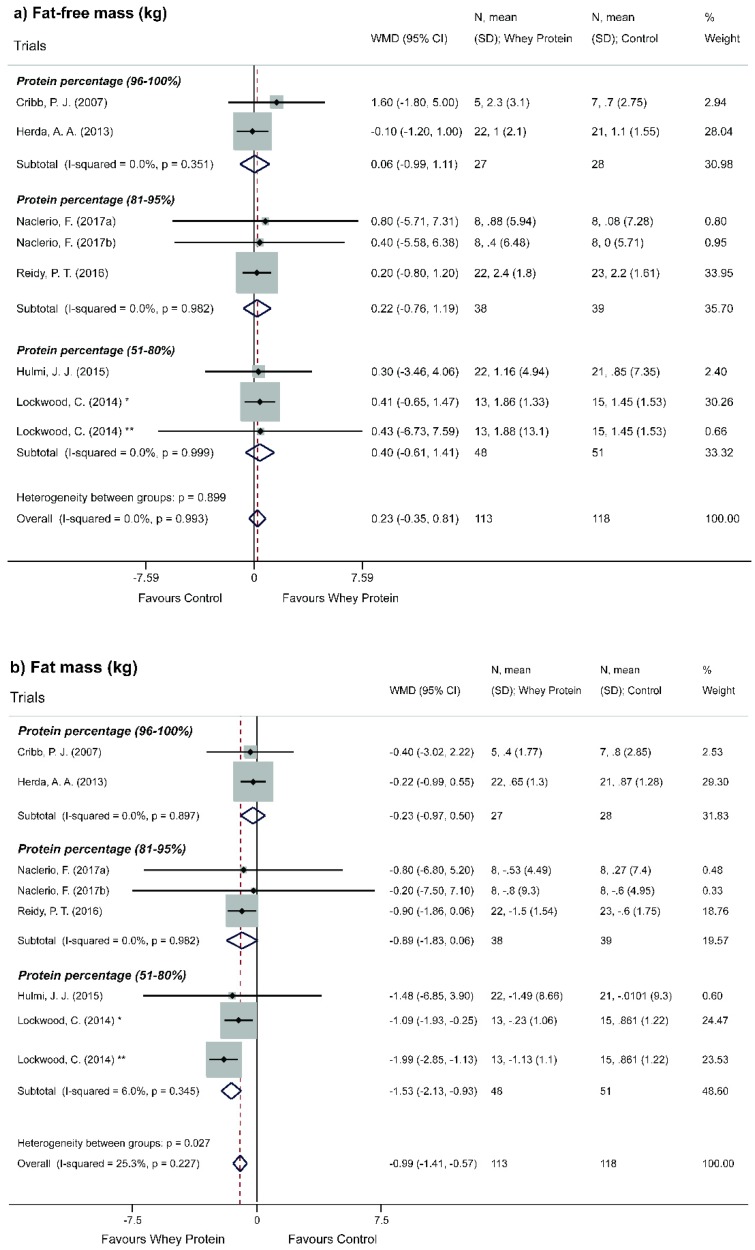
Meta-analysis of the effect of Whey Protein in subgroups for the protein percentage. (**a**) Fat-free mass. (**b**) Fat mass. * Whey protein concentrate. ** Whey protein hydrolyzed. After repeated unanswered contacts, one study [31] was omitted from this analysis due to not specifying the type of whey protein supplemented.

**Figure 6 nutrients-11-02047-f006:**
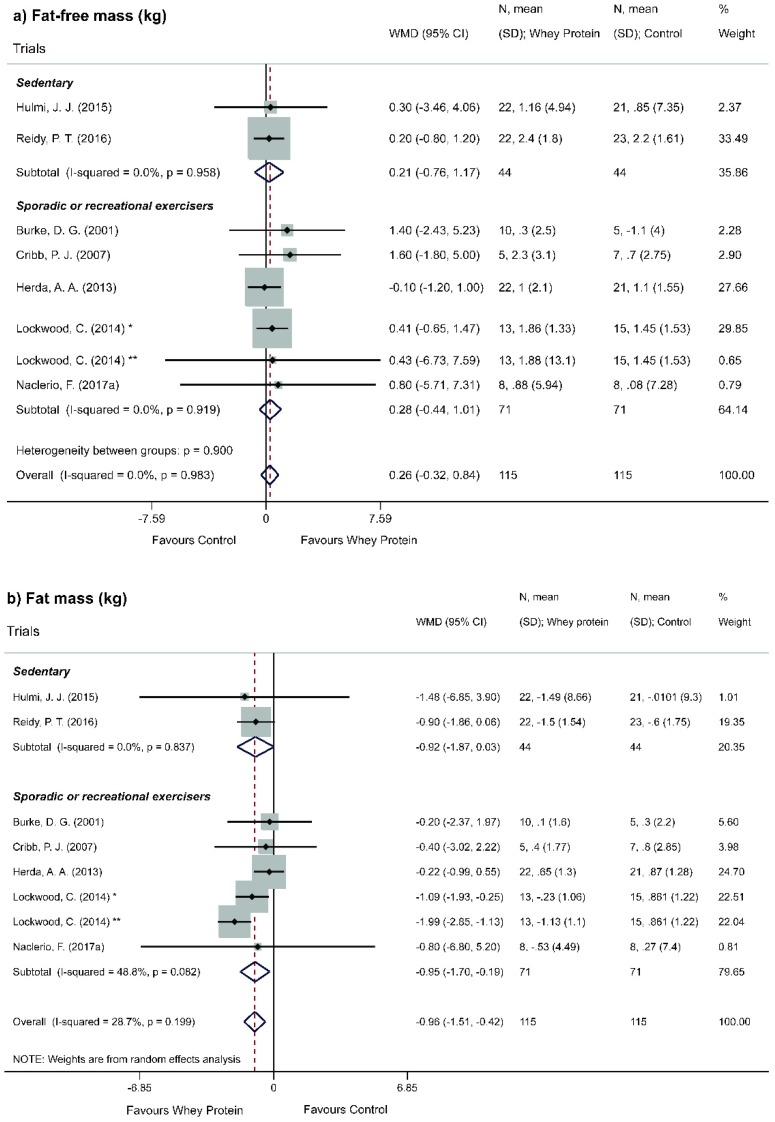
Meta-analysis of the effect of Whey Protein in subgroups for the level of physical activity. (**a**) Fat-free mass. (**b**) Fat mass. * Whey protein concentrate. ** Whey protein hydrolyzed. One RCT [37] was omitted from this analysis because it was the only study in which individuals were athletes.

**Table 1 nutrients-11-02047-t001:** PICO (Population, Intervention, Comparator, and Outcome) strategy and central seekers for regular studies.

Target	Description	Operator	Uniterms
Population	Healthy adults practicing physical activity	--	physical activity; physical exercise; training; exercise; coaching; resistance training; resistance exercise; gym; fitness; crossfit; weight lifting; non-sedentary; athlete.
		AND + NOT/AND NOT	child; pediatric; elder; aged; rat; mice.
Intervention	Whey protein supplementation (concentrated, hydrolyzed, and isolated)	AND	whey protein; whey; protein supplementation; whey supplementation; casein; casein supplementation; whey intake; casein intake; protein intake; whey concentrate; casein concentrate; concentrate protein; whey isolated; casein isolated; isolated protein; hydrolyzed whey; hydrolyzed casein; hydrolyzed protein; milk; milk protein; soy; soy protein.
Comparator	Randomized clinical trials	AND	controlled clinical trial; clinical trial; randomized clinical trial; group control; placebo; comparative study; cross-over; crossover; cross over; double-blind; factorial.
Outcome	Body composition related to hypertrophy	AND	muscle; mass gain; muscle gain; muscular gain; muscle strength; muscular strength; hypertrophy; lean mass; body composition; lean body mass; lean body tissue; fat-free mass; fat free mass; body weight; body mass; skeletal muscle.

**Table 2 nutrients-11-02047-t002:** Methodological characteristics of RCTs.

RCT	Country	Age (Years) (Minimum-Maximum)	Physical Activity Level *	Trainings per Week (Mean) **	Body Composition Measurement Method	Dose of WP (g·kg^−1^)	Nutritional Intervention	*n* (M:F)
Burke et al. [31]	Canada	18–31	Regular	4	DXA	1.2	WPI	10:0
							Maltodextrin	5:0
Cribb et al. [32]	Australia	18–31	Regular	4	DXA	1.28	WPI	5:0
							Non-specific carbohydrate	7:0
Herda et al. [33]	USA	18.5–23.9	Regular	3	Hydrostatic Weighing	0.25	WPC	22:0
							Maltodextrin	21:0
Hulmi et al. [34]	Finland	33.1–35.7	Sporadic	2.5	DXA	0.35	WPC	22:0
							Maltodextrin	21:0
Lockwood et al. [35]	USA	18–35	Regular	2	DXA	0.37	WPC	13:0
							Dextrose	15:0
							WPH	13:0
							Dextrose	15:0
Naclerio et al. [36]	USA	22–34	Regular	5	Plethysmography	0.26	WPI	8:0
							Non-specific carbohydrate	8:0
Naclerio et al. [37]	USA	36.4–54.2	Athlete	3	Plethysmography	0.24	WPI	8:0
							Non-specific carbohydrate	8:0
Reidy et al. [38]	USA	18–30	Sporadic	3	DXA	0.27	WPI	22:0
							Maltodextrin	23:0
Taylor et al. [39] ***	USA	18–24	Athlete	4	DXA	0.36	WPC	0:8
							Maltodextrin	0:6
Volek et al. [40] ***	USA	19.1–26.9	Sporadic	2.5	DXA	0.24	WPC	13:6
							Maltodextrin	13:9

* According to the criteria adopted for individuals prior to their entry into each study. ** During the application of the intervention protocol for all groups. *** Study included in qualitative analysis only. M: male. F: female. RCT: Randomized clinical trial. WP: whey protein. WPC: concentrate whey protein. WPH: hydrolyzed whey protein. WPI: isolated whey protein.

**Table 3 nutrients-11-02047-t003:** Basal moment, post-intervention, and delta of the values included in meta-analysis of lean and fat mass.

RCT		FFM						FM					
		WP			Placebo			WP			Placebo		
		PRE	POST	Δ	PRE	POST	Δ	PRE	POST	Δ	PRE	POST	Δ
Burke et al. [31]		62.30 ± 2.50	62.60 ± 2.50	+0.30	61.80 ± 4.00	60.70 ± 4.00	−1.10	13.90 ± 1.60	14.00 ± 1.60	+0.10	9.50 ± 2.20	9.80 ± 2.20	+0.30
Cribb et al. [32]		59.00 ± 3.20	61.30 ± 3.00	+2.30	62.30 ± 2.80	63.00 ± 2.70	+0.70	10.60 ± 1.90	11.00 ± 1.60	+0.40	13.2 ± 2.80	14.00 ± 2.90	+0.80
Herda et al. [33]		64.10 ± 2.10	65.10 ± 2.10	+1.00	64.60 ± 1.60	65.70 ± 1.50	+1.10	12.07 ± 0.80	12.72 ± 1.50	+0.65	12.06 ± 1.10	12.93 ± 1.40	+0.87
Hulmi et al. [34]		59.81 ± 4.72	60.96 ± 5.13	+1.15	60.02 ± 7.38	60.87 ± 7.30	+0.85	20.79 ± 8.87	19.31 ± 8.43	−1.48	18.40 ± 9.18	18.39 ± 9.41	−0.01
Lockwood et al. [35]	WPC	63.63 ± 1.34	65.50 ± 1.32	+1.86	59.10 ± 1.54	60.55 ± 1.51	+1.45	17.06 ± 1.08	16.83 ± 1.04	−0.22	9.46 ± 1.23	10.32 ± 1.19	+0.86
	WPH	60.51 ± 1.40	62.39 ± 13.74	+1.88	59.10 ± 1.54	60.55 ± 1.51	+1.45	18.94 ± 1.12	17.82 ± 1.08	−1.12	9.46 ± 1.23	10.32 ± 1.19	+0.86
Naclerio et al. [36]		66.10 ± 5.75	66.98 ± 6.12	+0.88	64.15 ± 7.28	64.23 ± 7.27	+0.08	11.94 ± 4.08	11.41 ± 4.81	−0.53	13.91 ± 7.48	14.18 ± 7.32	+0.27
Naclerio et al. [37]		65.40 ± 6.10	65.80 ± 6.80	+0.40	64.70 ± 5.90	64.70 ± 5.50	0.00	19.70 ± 8.40	18.9 ± 10.00	−0.80	15.40 ± 4.90	14.80 ± 5.00	−0.60
Reidy et al. [38]		57.60 ± 1.50	60.00 ± 2.00	+2.40	55.20 ± 1.50	57.40 ± 1.70	+2.20	20.50 ± 1.30	19.00 ± 1.70	−1.50	18.40 ± 1.70	17.80 ± 1.80	−0.60

Values expressed as mean (kg) ± standard deviation (kg). RCT: Randomized clinical trial. PRE: pre-intervention. POST: post-intervention. Δ: delta (final—initial value). WPC: concentrate whey protein. FFM: fat-free mass. FM: fat mass. WP: whey protein. WPH: hydrolyzed whey protein.

**Table 4 nutrients-11-02047-t004:** Physical activity protocol of the RCTs included in the meta-analysis.

RCT	Intensity	Volume			Total Log Duration (days)	Body Work	Outside Parallel Training
		Blocks	Duration (Days)	N° of Sets			
Burke et al. [31]	Adaptable to each participant’s progress based on RM	5	8	2	40	Whole body	Not mentioned
Cribb et al. [32]	70–95% RM	3	14–35	2–3	70	Whole body	Allowed, recorded, and analyzed
Herda et al. [33]	80% RM	2	7	2–3	56	Lower limbs	Allowed, but not registered
Hulmi et al. [34]	Adaptable to each participant’s progress based on the strength test (2–6 RM)	3	28	2–3	84	Whole body	Allowed for recreational activities, but not registered
Lockwood et al. [35]	5–10 RM	2 × 4	7	3	56	Whole body	Not mentioned
Naclerio et al. [36]	Adaptable to each participant’s progress based on RM	8	7	1	56	Whole body	Allowed in the protocol intervals, but only for recreational activities and not recorded
Naclerio et al. [37]	Adaptable to each participant’s progress based on the VT2 and HR_max_	4–6	14	3	70	Whole body	The usual training routine was used as an intervention protocol
Reidy et al. [38]	60–80% do RM	2	28–56	3–4	84	Whole body	Prohibited for force activity and allowed for recreational activities, but not recorded

RCT: Randomized clinical trial. RM: repetition maximum. VT2: ventilatory threshold 2. HR_max_: maximum heart rate.

**Table 5 nutrients-11-02047-t005:** Summary of Findings— Grading of Recommendations, Assessment, Development, and Evaluation (GRADE) tool.

Outcome			Participants (n) (WP: Placebo)	RCT (*n*)	WMD (95% CI)	Statistical Significance *	Quality of Evidence (GRADE)	Due to	Comments
FFM	Global		123:123	8	0.26 (−0.32, −0.83)	None	⊕⊕◯◯◯ Low	Imprecision	Range of effect possibly underestimated by random statistics and low number of events.
	WP type	WPC	57:57	3	0.17 (−0.58, 0.92)	None	⊕⊕◯◯◯ Low	Imprecision	Low sample size.
		WPI	53:51	6	0.38 (−0.53, 1.30)				
	Protein percentage (%)	96–100	27:28	2	0.06 (−0.99, 1.11)	None	⊕⊕◯◯◯ Low	Risk of bias	Only one study evaluated at low risk, others had consistent failures in randomization and in the selection of the reported result.
		81–95	38:39	3	0.22 (−0.76, 1.19)				
		51–80	48:51	2	0.40 (−0.61, 1.41)				
	Physical activity level	Sporadic or recreational	44:44	2	0.21 (−0.76, 1.17)	None	⊕⊕⊕◯◯ Moderate	Risk of bias	Presence of deviations from the intended interventions and inconsistencies in randomization.
		Regular	71:71	5	0.26 (−0.32, 0.84)				
FM	Global		123:123	8	−0.96 (−1.37, −0.55)	*p* < 0.001	⊕⊕⊕⊕◯ High	Risk of bias	Only one low-risk study.
	WP type	WPC	57:57	3	−0.63 (−1.19, −0.06)	*p* = 0.030	⊕⊕⊕⊕◯ High	Risk of bias	Only one low-risk study.
		WPI	110:108	5	−0.66 (−1.13, −0.20)	None			
	Protein percentage (%)	96–100	27:28	2	−0.23 (−0.97, 0.50)	None	⊕⊕⊕◯◯ Moderate	Imprecision	Range of effect possibly underestimated by random statistics and low number of events.
		81–95	38:39	3	−0.89 (−1.83, 0.06)	None			
		51–80	48:51	2	−1.53 (−2.13, −0.93)	*p* < 0.001			
		Global	113:118	7	−0.99 (−1.41, −0.57)	-	⊕◯◯◯◯ Very low	Indirect evidence	Metaregression not performed by the low number of studies (Figure 5b).
	Physical activity level	Sporadic or recreational	44:44	2	−0.92 (−1.87, 0.03)	None	⊕⊕⊕◯◯ Moderate	Risk of bias	Measure flaw (participants were not under the same restrictions to practice physical activity outside the study).
		Regular	71:71	5	−0.95 (−1.70, −0.19)	*p* = 0.014			

* None: value of *p* > 0.05. FFM: fat-free mass. FM fat mass. WP: whey protein. WPI: whey protein isolated. WPC: whey protein concentrate.

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
