# Peer review of "Comparative Meta-Analysis of the Effect of Concentrated, Hydrolyzed, and Isolated Whey Protein Supplementation on Body Composition of Physical Activity Practitioners"

_nutrients, 2019, doi:10.3390/nu11092047_

Round 1
Reviewer 1 Report
REVIEW REPORT
Manuscript: Comparative Meta-analysis of the Effects of Concentrated, Hydrolyzed, and Isolated Whey Protein Supplementation on Body Composition of Physical Activity Practitioners.
Dear Editor-in-chief,
Greetings!
Thanks for giving me the opportunity to review this paper which, I think, would be a good contribution to science. I read it several times with great interest as it is scientifically sound; the methodology used is well described and outcomes are well discussed, with a good writing style. I have a few minor comments.
Introduction section:- second page, 3rd paragraph: 'Conceptualized by the ... by this public.' This sentence should be corrected. The punctuation between 'visceral' and 'tissue' is not necessary I think; and the sentence is too long. Methods section:- Page 3, 1st paragraph: 'All methodological steps...each process.' This sentence should be corrected. It would be better to write 'two authors of this paper' instead of 'two members of the reviewers of this research'.
Author Response
Dear reviewer,
We are deeply grateful for the time spent reading our work. We are happy to know that, in your opinion, the results can contribute to science.
We carefully read all your comments and change the text to meet your requirements. The corrections can be found in the attached file.
In addition, considering your position, we requested professional assistance and an English-speaking author was invited to review the translated manuscript, as well as its scientific and methodological accuracy.
We hope the changes will be satisfactory, but if not, we are available for a rejoinder if you consider it necessary.
Thank you again for your interest in our manuscript.
Sincerely,
The authors.

Reviewer 2 Report
Dear authors,
Very interesting topic. However, please consider the following:
General comments:
1. English editing required.
2. Please consider to change "Lean Mass" in several sections of the manuscript, since your are mostly evaluating FFM data from DXA (7/10; qualitative analysis and 4/8; quantitative analysis). FFM or LM? Please read the following:
https://www.ncbi.nlm.nih.gov/pmc/articles/PMC6104103/
3. Avoid using first-person pronouns in the entire manuscript. Please make necessary corrections.
4. The authors only consider the difference between WPI and WPC with regards to protein content. However, several differences (e.g., protein fractions proportion, solubility, chemical behavior) are important to be declared in the introduction. Please refer to Morr, C. V., & Ha, E. Y. W. (1993). Whey protein concentrates and isolates: processing and functional properties. Critical Reviews in Food Science & Nutrition, 33(6), 431-476.
5. Please refer as "body mass" and not to "body weight" or "weight" (the last one is technically incorrect).
6. Nutritional timing for WP intake is not homogeneous in the RTCs. Please consider rewriting the conclusion about the effect of WP on LM/FFM, the readers tend to misinterpret.
7. From my perspective, the comparison of meta-analysis was pretentious considering the heterogeneity that was found overall in several RTC and the subsequent factors (energy expenditure, dietary intake, nutrient timing, etc.) - this was explained by the authors in "discussion" section.
8. The discussion section is cut by several mini-paragraphs. Authors should consider join many of them given that content analysis continues in some.
Specific changes:
Line 18: Replace "action" by effects. You should mention first FM and after LM (given the fact your results are in that order).
Line 19: especify muscle hypertrophy here.
Line 22-24: English editing. There better words for "submitted". Also, use the word "databases"
Line 25-27: Please express effects on FM and LM in terms of (xÌ… ± SD [CIs 95%], p value, effect size). Check grammar and punctuation.
Line 31-32: Erase meta-analysis since it is already in the title. Also, check "skeletal muscle" since your are mostly evaluting FFM data from DXA and not muscle mass directly.
Line 39-42: Please rewrite this paragraph. Also, erase the dot after osteoporosis in line 42.
Line 43-48: Even though protein content is important to distingish between WPI and WPC, there are other differences that should be mentioned. Please refer to Morr, C. V., & Ha, E. Y. W. (1993). Whey protein concentrates and isolates: processing and functional properties. Critical Reviews in Food Science & Nutrition, 33(6), 431-476.
Line 57-61: This is important. However, try making easier for the reader (check grammar and punctuation).
Line 63: Preferably refer to "body composition" instead of "anthropometrc markers".
Line 103: crossover studies
Line 117: This section is well described and performed. Good job.
Line 166: Use international units (kg not Kg). Same hereinafter.
Line 168-169: Use "dual energy X-ray absorptiometry" only. Replace DEXA by DXA.
Line 170-172: To the extend of this manuscript is ok. However, authors should be aware that experience following the ISAK guidelines for these measurements can decrease technical error (lower CV).
Line 295-297: Use linear form of the unit (raised to the -1). Use correct international system units (for kilogram is kg).
Line 300-302: This is important to highlight in the "limitations" section, since your analysis can have flaws.
Line 306 and Line 318: Replace "glycidic" by "carbohydrate".
Table 2: Use kg not Kg. Use F for female and M for male, instead of symbols.
Figure 4. Please check the values 0.26(-0.32, 0.83) for WMD (95% CI) in the variable lean mass (kg). The result is the same than Figure 3 (just to be sure). Please note the expected variation for WMD (95% CI) in Fat Mass between both figures (which is due to the withdraw of one of the samples of Lockwood et al.).
Table S5: Please keep the citation style of the journal in this table (authors' column).
Discussion: Authors should consider in the analysis that in order to increase LM / FFM and/or promote FM loss, it is important to program the nutrition strategy based on the energy balance. It is clear that the analyzed RTCs did not have the same nutritional intervention. Moreover, there are several paragraphs (Lines 533-557; Lines 602-618) with information that goes beyond the scope of the review; therefore, please be concise in the analysis and do not extend too much.
Line 442-444: Please modify. Currently, there is enough evidence to establish a recommendation of 0.4-0.5 g PRO / kg body mass after resistance training (Kersick et al., 2018; Maughan et al. 2018). The mentioned grams of WP per day by the authors is general belief or average intake of the population that should be erradicated. And even half ot this recommendation (20 g) is not enough for most of the population. Even though authors mentioned in the next paragraph some of these recommended values, I consider this make lost the attention of the idea and is not suitable for the scope.
See:
Kerksick, C. M., Wilborn, C. D., Roberts, M. D., Smith-Ryan, A., Kleiner, S. M., Jäger, R., ... & Greenwood, M. (2018). ISSN exercise & sports nutrition review update: research & recommendations. Journal of the International Society of Sports Nutrition, 15(1), 38.
Maughan, R. J., Burke, L. M., Dvorak, J., Larson-Meyer, D. E., Peeling, P., Phillips, S. M., ... & Meeusen, R. (2018). IOC consensus statement: dietary supplements and the high-performance athlete. International journal of sport nutrition and exercise metabolism, 28(2), 104-125.
Line 472-476: This clarification is important. However, remember to replace "body mass" instead of "weight".
Line 477-478: Replace "moment of ingestion" by "nutrient timing".
Line 485-487: Authors must support this based on their results, because the analyzed RTCs (conditions, methods, results and statistical effect) are different to those from Schoenfeld et al. (2013).
Line 495: The analysis by cutoff points for protein content has flaws, because data and RTCs are not homogeneous. The interventions were different to establish an effect based on different protein content.
Line 613-614: Use abbreviation MPS (this was included previously).
Line 658 (Section 4.2.): Please change all abbreviation that have been mentioned before (LM, FM, etc.)
Line 858-868 (Conclusions section): Please do not use figurative language. Try making conclusions more realistic according to your results and limitations.
Author Response
Dear reviewer,
We consider that the article has greatly benefited in its scientific and linguistic character with your considerations. We hope that the changes have met all the requirements, but otherwise we will be happy to improve the text again according to your demands.
We would like to thank you for the time invested in reading our work and also for your efforts in improving it. Your contributions have substantially enhanced the results interpretation.
The amendments are available in the attached file and we appreciate the interest in our manuscript.
Sincerely,
The authors.

Round 2
Reviewer 2 Report
Dear authors,
Thanks for taking the time for answering.
Just few final comments:
In abstract section, if you use an abbreviation (e.g., WMD) you should mention what it stands for. Change the expression "better evidence" and remark that this is regarding to FFM gain, because the effects on FM were revealed by your analysis.
Use FFM and FM after you used the abbreviation first time (e.g., the statistical analysis section and several others use fat-free mass and fat mass instead of the abbreviation). Check hereinafter.
Author Response
Dear reviewer,
We thank you again for the time dedicated to our manuscript. Working on the notes suggested by your reading was a gratifying experience for our research group.
The changes were made and are described in the attached file.
Thank you again.
Greetings,
The authors.
